# Factors Affecting Risk Attitude of Rice Farmers: Evidence from Vietnam's Mekong Delta

Khuu Thi Phuong Dong 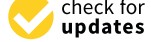, Phan Dinh Khoi *, Phan Hong Nhung, Nguyen Thanh Binh and Tran Thi Hanh Phuc

School of Economics, Can Tho University, Can Tho City 900000, Vietnam; ktpdong@ctu.edu.vn (K.T.P.D.);
nhungm2718023@gstudent.ctu.edu.vn (P.H.N.); nthanhbinh@ctu.edu.vn (N.T.B.); tthphuc@ctu.edu.vn (T.T.H.P.)
* Correspondence: pdkhoi@ctu.edu.vn

**Abstract:** Agricultural production accounts for 64.2% of the Vietnam's Mekong Delta. However, this sector has to face damage risks, especially from the natural disasters, such as flood, drought, severe soil salinity, pests, and erosion, which might factor into the farmers' risk attitude and their decision-making relative to investment in production activities. This study analyzes the factors influencing the risk attitudes of the rice farmers, based on evidence from the Vietnamese Mekong Delta. The data were collected through face-to-face interviews and experimental games with 145 rice farmers. An ordered probit regression model was applied to estimate how the factors affected the rice farmers' risk attitudes. The risk-neutral farmers comprised 53.72% of farmers in the survey, while 31.72% and 15.15% were risk-preferred and risk-averse farmers. The study results indicated that age, number of rice crops per year, household assets, income from rice production, and credit accessibility were the main factors affecting the farmers' risk attitudes. The results suggest that the financial incentives' policies to compensate for losses in uncertain conditions and increase the household income, diversification of income sources, and improving the accessibility of formal credit might be useful to increase farmers' willingness to accept the risks of investing in better profitability projects and gaining a higher income.

**Keywords:** risk attitudes; rice production; ordered probit regression model

## 1. Introduction

Vietnam's Mekong Delta region covers a total area of approximately 39.747 km$^2$, accounting for 12.25% of Vietnam's total area. Agricultural production accounts for 64.2% of the region, and rice farming is the largest production area. Recently, several agricultural restructuring projects have been initiated and have brought evident benefits for regions, such as increases in farmers' incomes. However, the region continues to face enormous farming, institutional, personal, financial, market, and climate change risks (flood, drought, severe soil salinity, pests, and erosion). Additionally, fluctuations in income and outcomes reveal the necessity of agricultural restructuring in the region.

Many studies have been conducted to evaluate these risks and their effects on agricultural production. Ninh (2013) stated that risk is the sum of contingency events that occur during most human activities. Risks can result in losses, but can also bring benefits and opportunities. Knight (1921) indicated that each individual can make different speculations with different possibilities and, hence, have different decisions. Personal decision-making in a risky situation can be considered as a risk attitude. This depends on the personal evaluation of the perceived individual's risk (Belaid and Miller 1987).

The occurrence of uncertainties and risks (e.g., natural disasters, fluctuation of markets, institutional risks, and social instability) has encouraged their inclusion in farmers' decision-making related to investment in production activities (Mendola 2007). Notably, farmers are considered risk averse as they frequently lack financial resources and cannot manage the negative effects on their production (Taylor and Adelman 2003). Therefore, farmers' risk attitudes significantly influence their investment decisions.

Consequently, farmers' attitudes toward risk (i.e., risk attitudes) are reflected by careful investment decision-making in the interest of protecting their families. This results in farmers choosing safe investment strategies to minimize risks (Nurul and Hasanuzzaman 2013). Therefore, farmers are expected to reduce their income, as they may forgo higher potential return investments that offer greater risk. They might then become poorer, owing to reduced risk tolerance. Therefore, examining farmers' risk attitudes is critical to ensure that they can maximize utility through their choices under uncertain conditions.

Expected utility theory, which was first proposed by Deming et al. (1944), clarifies the relationship between farmers' expectations and their investment decision-making under uncertain situations (Coulhon and Mongin 1989). Under risk conditions, decision makers tend to choose and make investment decisions by comparing the expected utility values with total utility values generated from decisions made, multiplied by the probabilities of risk occurrence (Mongin 1997; Chai et al. 2021).

Various empirical studies have addressed the conflict between risk and production choice. Rosenzweig and Binswanger (1993) estimated the impact of risk based on the measurement of precipitation variability on farmers' investment portfolios. The results suggest that weather risk is key in low production efficiency. The farmers operating in a risky environment select portfolios with less risk (e.g., less sensitive to precipitation), but also probably less profitable. Morduch (1994) found similar evidence from Indian farmers, who face a higher probability of losing the asset value owing to damage risks and are more likely to prefer conservative strategies with lower returns than risky strategies with potentially high returns.

Risk attitudes differ among farmers because it depends on different factors, such as psychological characteristics (i.e., gender, ages, income, occupation, etc.), social conditions, natural conditions, and individual perceptions of risks (Slovic 1992; Yilmazer and Lyons 2010; Giannikos and Korkou 2022). The farmers' risk attitudes were divided into three groups: risk averse, risk neutral, and risk preferred group (Ellis 1998). The farmers with risk-averse attitudes prefer the options that result in income security or well-being, even though they may know that the benefits can be much lower than risky options. Conversely, individuals with risk-preferred attitudes prefer options with a high chance of yielding, although the probability of failure may be high.

However, Roumasset (1976) argued that farmers' decision-making behavior is not only affected by risk preferences but also by being proactive in how production is organized to increase risk tolerance. Therefore, farmers' risk attitudes can be considered as a factor influencing their choices in their production activities (Eswaran and Kotwal 1986; Morduch 1994). Therefore, studying farmers' risk attitudes allow for the development of a risk-sharing mechanism among stakeholders, including farmers, local authorities, and suppliers who provide support services, to minimize the risk of damage.

Essentially, the studies on production decisions under risk conditions are key for the development of the platform (i.e., agricultural insurance services), to manage and respond to damage control strategies (Shawn et al. 2013; Clarke 2011). Those studies mostly demonstrated that willingness to pay for agricultural insurance services increases with risk aversion. Therefore, measuring farmers' risk attitudes and identifying the factors influencing risk attitudes are necessary to design the mechanism of agricultural insurance contracts, including rice crop insurance. Similarly, Varangis et al. (2002) found that formal risk-sharing mechanisms (i.e., insurance, trading in commodity futures, or options), which have been fully established in developed countries, are important for risk management procedures in the agricultural sector. However, these mechanisms are unavailable or are only at the starting point in developing countries (i.e., Vietnam).

Considering the scientific perspectives, several studies relating to the agricultural insurance sector were conducted in Vietnam (Khoi et al. 2017; Huy and Khoi 2015; Son 2012; Vandeveer 2001). These studies focused on describing the role, development procedures, and limitations of Vietnam's agricultural insurance services. Studies on risk attitudes and risk-sharing mechanisms of the stakeholders in agricultural insurance in general,

and rice crop insurance in particular, have not yet been conducted. This study aimed to determine the factors affecting rice farmers' risk attitudes in the Vietnam's Mekong Delta, with evidence from the two provinces of An Giang and Hau Giang. The two provinces were selected for the case study owing to rice production being the main source of income in the two provinces. In 2020, Hau Giang province had about 196.1 thousand hectares of rice production, the output was 1258.4 thousand tons, while the average yield was about 642 kgs/ha; in contrast, An Giang province has about 626.2 thousand hectares of rice production, the output was 3916.8 thousand tons, while the average yield was about 625 kgs/ha. Additionally, the rice processing activities are conducted in these two provinces, creating jobs for many people in rural areas and contributing to stabilizing social security. However, the rice production activities in these provinces are in the same situation as the whole of the Vietnam's Mekong Delta region, since the farmers have to deal with the negative effects of various risks from weather damage on the production. This study on the risk attitude of rice-farmers, thus, is expected to provide an important scientific basis for proposing relevant policy implications to help farmers in the Vietnam's Mekong Delta region become bold in their investment decisions, scale up, and reduce the impacts of risks on production activities.

## 2. Methodology

### 2.1. Data Collection

The secondary data were collected from the General Statistics Office of Vietnam, Department of Agriculture and Rural Development of Hau Giang and An Giang provinces, reports on socioeconomic development of the region and provinces, and relevant articles and scientific reports.

The primary data were collected in June 2020 using stratified sampling combined with random sampling. The stratification criteria included production site, type, and scale. A total of 145 rice farmers, including 101 and 44 rice farmers in Hau Giang and An Giang Provinces, respectively, were selected for the interviews.

The experimental method, with the actual bonus being paid to the participants based on a "multiple price list", created by Binswanger (1980) and Dohmen et al. (2011), was used to conduct the investigation into the rice farmers' risk attitude, including risk averse, risk neutral, and risk preferred. The participants were asked to choose between the proposed bids with different risk levels. The participants' risk attitudes were considered to be based on their choice. The respondents selected to participate in the survey were those who directly made the decisions and participated in the household rice production activities.

The survey procedure was carried out as follows:

First, the interviewer asks the players to choose from different risk situations within the range of the designed game so that their expected payoff can be achieved. Each respondent was required to play through three rounds of the game of chance and be rewarded with real money. Round 1 determines the farmer's risk attitude and rounds 2 and 3 determine the reasonableness and accuracy of the previous choice. The implementation of multiple rounds is intended to measure players' risk attitudes more accurately by matching results between rounds. The game is described as follows:

1. Step 1: The interviewer explains the rules and bonuses to the players. Accordingly, the base bonus of the game will be 50,000 VND, approximately two hours of the average payment that a local farmer is paid for working;

2. Step 2: This step includes the following three tiers:

■ Round 1: This round is intended to determine the player's risk attitude. The interviewer gives player 10 face-down cards, of which five are black and five are red. Players were required to choose only one of three options, as follows:

✔ Option A: Do not participate in the game and receive 50,000 VND immediately;

✔      Option B: Participate in the game and choose one random card; if the draw is a red card, player will receive 100,000 VND; if player draws a black card, player will receive nothing;

✔      Option C: Respondents think that choice A or B makes no difference.

A risk-averse player is expected to choose option A because by choosing A, the player receives a sum of money without facing any risk. A risk-preferred player will choose B as this is a risky choice, and the player's reward is determined by the choice of whether to participate in the game with a probability of winning or losing, if participating in the game is 50/50 and the standard deviation of the reward is 0. The risk-neutral player chooses option C.

To confirm risk attitude, the player was then asked to play Round 2, as follows:

▪      Round 2: The 10 cards in turn 1 will be replaced with 10 cards, consisting of three black cards and seven red cards. The players have two options as follows:

✔      Option 1: Do not participate in the game and receive 50,000 VND immediately;

✔      Option 2: Participate in the game and choose one random card; if the draw is a red card, player will receive 100,000 VND; if the player draws a black card, the player will receive nothing.

In this round, option number 3 "Option C: two options A and B have no difference" was removed so that the farmer's risk attitude can be accurately determined. Additionally, the probability of receiving 100,000 VND if participating in the game increased to 70%. Thus, the player who chooses option A is risk averse. The risk-preferred player chooses option B. Because the expected value has increased, the risk-neutral group will be more inclined to choose option B than option A.

▪      Round 3: The 10 cards in turn 1 will be replaced with 10 cards, consisting of one black card and nine red cards. players have two options, as follows:

✔      Option 1: Do not participate in the game and receive 50,000 VND immediately;

✔      Option 2: Participate in the game and choose one random card; if the draw is a red card, player will receive 100,000 VND; if the player draws a black card, player will receive nothing.

In this round, the probability of receiving 100,000 VND if participating in the game has increased to 90%. Thus, the player selecting option A is definitely risk averse. A risk-preferred or risk-neutral player will choose option B, as the expected value of this turn has increased to almost absolute.

After participating in the games, the respondents were asked to answer a questionnaire to collect information to estimate the factors affecting their risk attitude. This questionnaire comprised three sections. Section 1 dealt with information on the sociodemographic characteristics of respondents and households. Section 2 contained information related to the financial characteristics of the household. Finally, Section 3 included information related to production activities and the risks encountered in the production activities of rice farmers.

### 2.2. Data Analysis

In this study, rice farmers' risk attitudes included three groups: risk averse, risk neutral, and risk preferred. Thus, an ordered probit regression model was applied to estimate the factors' effect on rice farmers' risk attitudes, as suggested by Dohmen et al. (2011) and Lune and Berg (2017). $Y_i^*$ presented in Equation (1) is an unobserved independent variable that has a linear relationship with the characteristics of rice farmer $X_i$ ($X_i$ is the observed independent variable). Let $Y_i$ be the observed risk according to risk preferred, neutrality, and aversion (see Equation (2)). $Y_i$ takes the values 0, 1, and 2, respectively. The ordered probit regression model is as follows:

$$Y_i^* = X_i\beta + \varepsilon_i, \text{ with } i = 1 \dots N \quad (1)$$

$$Y_i = \begin{cases} 0 \; \forall \; Y_i^* \leq \mu_1 \\ 1 \; \forall \; \mu_1 < Y_i^* \leq \mu_2 \\ 2 \; \forall \; Y_i^* > \mu_2 \end{cases}, \text{ with } \mu_1, \mu_2 \text{ is the threshold for accepting risk.} \qquad (2)$$

The probability to determine the household's risk attitude is as follows (see Equation (3)):

$$\begin{aligned} Pr(Y_i = 0) &= \Phi(\mu_i - X_i\beta) \\ Pr(Y_i = 1) &= \Phi(\mu_2 - X_i\beta) - \Phi(\mu_1 - X_i\beta) \\ Pr(Y_i = 2) &= 1 - \Phi(\mu_2 - X_i\beta) \end{aligned} \qquad (3)$$

where, $\Phi$ is the cumulative probability distribution function of a normal distribution. Coefficient $\beta$ was estimated using the maximum likelihood estimation (MLE) method at a significance level of 10% ($P < 0.1$). However, coefficient $\beta$ does not directly present the relationship between the factors affecting risk attitude. Hence, the marginal effect, which is calculated according to the platform suggested by Wooldridge (2002) and Bartus and Roodman (2014), is used to explain the effect of the observed characteristics of households with their risk attitudes.

The factors affecting the risk attitude of rice farmers in An Giang province and Hau Giang province were synthesized from relevant studies. In this study, a group of factors on financial characteristics, including assets, income, savings, and access to loan capital, are introduced into the model to determine the impact of characteristics on financial resources on rice farmers' risk attitudes. Lipton (1968) explained that, under uncertain conditions, income will affect the farmers' risk aversion, and they can only accept risks when the potential for an increase in income is evident. The farmers financial resources have a direct influence on their risk attitudes (Binswanger 1980). Farmers with sufficient financial resources prioritize investment in production activities, expand when necessary to maximize profits, and often have a neutral risk attitude (Antle 1989). Conversely, the households that have fewer resources, and particularly, if in the context of imperfect credit markets, poor farmers without collateral are often unable to cope with risks (Eswaran and Kotwal 1986). Therefore, the farmers will choose low-risk investment portfolios or those that produce corresponding returns with minimal uncertainty to minimize risk (Morduch 1994).

The frequency of risk occurrence and rice yield were included in the model to determine the impact of these factors on farmers' risk attitudes (Rosenzweig and Binswanger 1993). Additionally, the number of rice production crops (per year) was also included in the model to assess the impact on farmers' risk attitudes. In Vietnam, research by Diep et al. (2015) has demonstrated that the group of households producing three rice crops per year will face higher risks of weather and soil salinity than the group of households producing a rice-crop rotation, such as two rice–one crop, or one rice–two crops. Therefore, the farmers producing three rice crops in a monoculture model are vulnerable to risks in production and tend to be more averse to risks than other groups of farmers.

Willingness to participate in crop insurance products increases with risk aversion, and risk attitude is key in designing a crop insurance contract (Clarke 2011). In this study, farmers' willingness to participate in rice insurance products was included in the model to examine this factor's influence on their risk attitudes.

Sociological factors, including age, expertise in rice production, and production experience, were also included in the model to determine their influence on rice farmers' risk attitudes (Ullah et al. 2015). The family member variable was also included in the model to examine the influence of the number of family members on farmers' risk attitudes. Morduch (1995) argues that farmers' investment decisions for production can be influenced by other household members. Simultaneously, financial pressure from dependent members in the household also increases farmers' risk aversion in investment decisions. Table 1 presents the interpretation of the independent variables used in the ordered probit regression model.

**Table 1.** Description of variables in the ordered probit regression model.

| Variables | Description | Measurement |
|---|---|---|
| Independent variables | | |
| | Risk attitude | 0: Risk preferred<br>1: Risk neutral<br>2: Risk averse |
| Dependent variables | | |
| X1 | Age | Age of respondent |
| X2 | Training level in agricultural production | Obtain a value of 1 if the respondent has not been trained in agricultural production; obtain a value of 2 if the respondent has undertaken short training; obtain a value of 3 if the respondent has an elementary degree in agriculture; obtain a value of 4 if the respondent has an agricultural intermediate degree or higher |
| X3 | Numbers of household member | Number of members in the household |
| X4 | Asset | Total asset value of the household |
| X5 | Loan capital | Obtain a value of 1 if the respondent has an agricultural production loan; obtain a value of 0 if the respondent does not take a loan |
| X6 | Saving | Obtain a value of 1 if the respondent has savings from his income; obtain a value of 0 if the respondent does not save |
| X7 | Number of rice crops | Number of rice crops in 2020 |
| X8 | Cost | Average cost of rice production per hectare in 2020 (million VND) |
| X9 | Rice yield | Average rice yield in 2020 |
| X10 | Income from rice production activities | Income from rice production activities of households in 2020 (million VND) |
| X11 | Income from other production activities | Income from other production activities of the household in 2020 (million VND) |
| X12 | Average frequency of risks | Average number of risks in rice production of the household from 2018 to 2020 (times/year) |
| X13 | Willingness to participate in agricultural insurance | Obtain a value of 1 if the respondent chose to be willing to participate in agricultural insurance, obtain a value of 0 if the respondent chose not to participate in agricultural insurance |

## 3. Results

*3.1. Measuring Risk Attitude of Rice Farmers in the Mekong Delta, Vietnam: Evidence from Hau Giang and An Giang Provinces*

Table 2 shows a summary of the results of the experiment to measure the risk attitudes of rice farmers.

**Table 2.** Risk attitude of rice farmers in Hau Giang and An Giang provinces.

| No. | Options | Hau Giang Province | | An Giang Province | | Whole Samples | |
|---|---|---|---|---|---|---|---|
| | | Freq. | Ratio (%) | Freq. | Ratio (%) | Freq. | Ratio (%) |
| 1 | Round 1 | | | | | | |
| | Option A | 34 | 33.66 | 15 | 34.09 | 49 | 33.79 |
| | Option B | 23 | 22.77 | 11 | 25 | 34 | 23.45 |
| | Option C | 44 | 43.56 | 18 | 40.91 | 62 | 42.76 |
| 2 | Round 2 | | | | | | |
| | Option A | 45 | 44.55 | 22 | 50 | 67 | 46.21 |
| | Option B | 56 | 55.45 | 22 | 50 | 78 | 53.79 |
| 3 | Round 3 | | | | | | |
| | Option A | 15 | 14.85 | 7 | 15.91 | 22 | 15.17 |
| | Option B | 86 | 85.15 | 37 | 84.09 | 123 | 84.83 |

Note: Authors calculated from survey data collected in the Hau Giang and An Giang provinces.

In the first round, 34/145 households (23.45%) chose option B (Table 2). In other words, the number of players choosing risky options has the lowest ratio among the options. Essentially, in the first round, the percentage of risk-preferred respondents was lowest among the groups. In the second round, the acceptance rate of participating in the game increased to 78/145 households (accounting for 53.79%), which may be because the probability of winning when participating in the game increased to 70%. In the third round, the probability that the player will win if the player chooses the risky option is 90%.

The experimental results showed that the number of players choosing risky options increased to 123/145 (84.83%). Hence, the experimental results of all three rounds demonstrated that the farmers participating in the survey tended to be risk neutral toward risky game choices. The experimental results in the third round showed that, when winning rate was increased to 90%, approximately 15% of the farmers participating in the experiment chose the safe option (did not participate in the game). Hence, farmers still tended to make decisions with certain outcomes, and, hence, are not risk preferred. This result is consistent with those of De and Thong (2020), Samuel et al. (2012), Sulewski and Kłoczko-Gajewska (2014), Ullah et al. (2015), Fahad et al. (2018), and Komarek et al. (2020). The risk-averse attitude of farmers might prevent investment in risky projects and then probably lose the chance to maximize profit from rice farmers' investments. This might have a negative effect on their income, especially for poor farmers.

### 3.2. Factors Affecting to Risk Attitude of Rice Farmers in the Mekong Delta, Vietnam: Evidence from Hau Giang and An Giang Provinces

An ordered probit regression model was used to determine the factors that affect rice farmers' risk attitude toward Hau Giang and An Giang provinces. Table 3 presents the results of the regression analysis.

**Table 3.** Results of ordered probit regression model.

| Variables | Symbols | Coefficients | Marginal Effects | | |
|---|---|---|---|---|---|
| | | | **Risk-Preferred** | **Risk-Neutral** | **Risk-Averse** |
| Age | X1 | 0.264 ** | −0.009 ** | 0.004 ** | 0.005 ** |
| Training level in agricultural production | X2 | 0.149 | −0.051 | 0.022 | 0.030 |
| Numbers of household member | X3 | −0.098 | 0.034 | −0.014 | −0.019 |
| Asset | X4 | 0.001 * | −0.001 * | 0.001 * | 0.001 * |
| Loan capital | X5 | 0.917 ** | −0.315 ** | 0.134 ** | 0.181 ** |
| Saving | X6 | 0.201 | −0.069 | 0.029 | 0.040 |
| Number of rice crops | X7 | 0.072 | 0.002 | −0.001 | −0.001 |
| Cost | X8 | −0.007 ** | 0.001 ** | −0.001 ** | −0.001 ** |
| Rice yield | X9 | −0.002 | 0.308 | −0.131 | −0.177 |
| Income from rice production activities | X10 | −0.896 ** | 0.101 ** | −0.043 ** | −0.058 ** |
| Income from other production activities | X11 | −0.293 | −0.006 | 0.002 | 0.003 |
| Average frequency of risks | X12 | 0.016 | −0.117 | 0.050 | 0.0674 |
| Willingness to participate in agricultural insurance | X13 | 0.341 | −0.025 | 0.011 | 0.014 |
| Total observation | | 145 | | | |
| Pseudo R2 | | 0.114 | | | |
| Prob > Chi2 | | 0.002 | | | |
| LR value (Likelihood Ratio) | | 32.50 | | | |

Note: Authors estimated from survey data collected in the Hau Giang and An Giang provinces. **, * indicates the level of significane at 5%, and 10%.

The correlation coefficient of the ordered probit regression model revealed that the respondents' age, assets, number of rice crops in the year, income from rice production, and use of rice production loans affected the rice farmers' risk attitudes ($P < 0.1$). The results of the marginal effects for each group of risk attitudes, including the risk-preferred, risk-neutral, and risk-averse farmers (Table 3), present the following.

For the risk-preferred group, if age increases by one unit, the probability of farmers switching from the risk-preferred group to the risk-neutral group decreases by 0.9 percentage points, while the other factors are constant ($P < 0.1$). Similarly, the higher the household asset value, the lower the probability of the farmer switching to the risk-neutral group decreases by 0.01 percentage points, all factors held constant ($P < 0.1$). If the number of rice production crops in the year increases by one crop, the probability that farmers switched from a risk-preferred to a risk-neutral group decreases by 31.5 percentage points, all factors held constant. An increase in income from rice production by one unit will increase the probability that farmers switch from a risk-preferred to a risk-neutral group by 0.24 percentage points ($P < 0.1$). Meanwhile, the households using loans for rice production will increase the probability that the farmers switch from a risk-preferred to a risk-neutral group by 30.8 percentage points ($P < 0.1$), when other factors remain unchanged.

For the group of farmers with a risk-neutral attitude, when other factors remain unchanged, if age increases by one unit, the probability of farmers switching from the group with a risk-neutral attitude to the group with high risk aversion increased by 0.39 percentage points ($P < 0.1$). If the number of rice production crops in the year increases by one crop, the probability that farmers switch from the group with a risk-neutral attitude to the group with a risk-averse attitude increases by 13.4% percentage points ($P < 0.1$). If income from rice production decreases by one unit, the probability that farmers switch from the group with a risk-neutral attitude to the group with a risk-averse attitude increases by 0.1 percentage points ($P < 0.1$). In the households using rice production loans, the probability that the farmers with a risk-neutral attitude switch to a group with a risk-averse attitude is increased by 13.1 percentage points ($P < 0.1$).

For the groups of farmers with a risk-averse attitude, if age decreases by one unit, the probability of switching from a group with a risk-averse attitude to a group with a risk-neutral attitude increases by 0.52 percentage points ($P < 0.1$). The marginal coefficient of the asset value of the household has a negative value. This means that if the household's assets decrease by one unit, the probability that the farmer will switch from the group with a risk-averse attitude to that with a risk-neutral attitude increases by 0.7 percentage points ($P < 0.1$). The regression results also demonstrated that for farmers with a decrease of one in the number of rice crops produced in a year, the probability that they will switch from the risk-averse group to the group with a neutral risk attitude increases by 18.1 percentage points, respectively ($P < 0.1$). If a household's income from rice production increases by one unit, the probability that farmers will switch from a group with a risk-averse attitude to a group with a neutral risk attitude increases by 0.14 percentage points. Our analysis results also demonstrate that the use of loans for rice production increases the probability that farmers will switch from a group with a risk-averse attitude to a group with a risk-neutral attitude by 17.7 percentage points.

## 4. Discussions

Overall, the regression analysis results in the ordered probit regression model identified the factors affecting the risk attitudes of the rice farmers in the Mekong Delta, with experimental evidence from the Hau Giang and An Giang provinces. The respondents' age was positively correlated with farmers' risk aversion ($P < 0.1$). The study results were consistent with those of Ullah et al. (2015) and Fahad et al. (2018). Hence, the probability that farmers will change their attitude from risk averse to risk neutral will increase when they are younger, if all of the other factors remain unchanged. This might be explained by the fact that younger farmers might have a higher motivation to earn a higher income. Hence, they might accept the challenges of investing in risky production activities, compared to older farmers.

In contrast, Nigist (2007) indicated the opposite results, such that the characteristics of the household head (e.g., age and education level) did not affect farmers' risk attitudes. Nigist (2007) agreed that risk attitude depends on a farmer's asset value; however, farmers who have fewer assets tend to be risk averse. Poorer farmers lack the financial resources to

respond to potential calamities in their life and production processes. As mentioned in the introduction, farmers' risk aversion attitudes might lead them to decline high profitability investments and challenge the poverty trap.

Regarding the groups of financial factors, asset value had a positive influence on rice farmers' risk aversion. The results of this study are not consistent with those of Binswanger (1980), Khoi et al. (2017), Liu (2013), and Kiet and Phat (2019), who reported that farmers with a lower value of assets tended to be risk averse as they lacked the financial resources to respond to potential calamities in their life and production process. However, the results of this study indicate that the probability of changing a farmer's attitude from risk averse to risk neutral and risk preferred increases when the asset value is reduced by one unit. This means that the higher the value of assets that farmers have, the higher their risk aversion. This result corresponds with evidence from Indian rice farmers found by Morduch (1994). Accordingly, rice farmers are generally risk averse. Moreover, they disagreed with the cultivation of high-productivity seeds, as the probability of the occurrence of uncertainty and risks from natural conditions when using this kind of seed might be higher. However, Morduch (1995) also found that farmers who have a higher asset value prefer to select safe projects with lower potential risks, as they do not want to trade-off recent prosperity.

Conversely, an increase in income from rice production will help farmers reduce their risk-averse attitude ($P < 0.1$). An increase in income might call for farmers' confidence in accepting new challenges (Lipton 1968). Therefore, willingness to tolerate risks and uncertainties in the production process is estimated to be higher. Additionally, the group of farmers using rice production loans also has a lower risk-averse attitude than those who did not ($P < 0.1$). The results of this study were consistent with those of Kiet and Phat (2019) and De and Thong (2021). These results can be explained by the fact that loans increase additional costs and create a financial burden on farmers. Therefore, farmers who are willing to use loans for production might be considered risk preferred. This suggests that accessibility to credits with a reasonable interest rate might enhance farmers' willingness to accept a higher probability of risk. This, then, might help farmers gain higher profitability projects, which might contain higher potential risks.

Thus, financial conditions have been considered as significant factors affecting farmers' risk attitudes. The farmers who participated in the experimental survey might have declined the high profitability investment to guarantee their asset value in damaged situations and risks. However, the effects of willingness to participate in rice-crop insurance on farmers' risk attitudes have not been confirmed ($P < 0.1$). This might be because rice-crop insurance is still being piloted in Vietnam, as mentioned in the introduction. Therefore, farmers do not perceive the significant benefits of these programs. However, recent results on the negative effects of asset value on risk tolerance have suggested that alternatives for minimizing the damages assure that the value of assets might encourage farmers to accept a higher probability of risks and uncertainties. Innes (2003) indicated that the government might provide subsidies to compensate for farmers' losses. However, this might impose a long-term burden on public budgets. Accordingly, rice crop insurance might be considered to support the compensation of farmers' losses due to risks (Jin et al. 2017).

Additionally, an increase in income and convenience in credit accessibility might enhance farmers' risk tolerance, suggesting that incentive policies in financial resources from the national authorities might enhance farmers' willingness to invest and expand their investment. This creates the opportunity to earn higher income and reduce the poverty of rice farmers, especially of poor farmers in rural areas.

Conversely, the group of farmers with less than one rice crop per year will have a less risk-averse attitude ($P < 0.1$) when other factors are constant. This result is consistent with Diep et al. (2015) and Url et al. (2018). This remains consistent with the summary of the data collected in this study. The households with fewer rice crops may switch to other production activities. This means that households willing to switch in production activities might have a lower risk aversion than those unwilling to switch. This suggests

that diversification of income sources rather than dependence on income from rice farming might create changes in the risk aversion attitudes of rice farmers (Belaid and Miller 1987).

Regression analysis also did not find an impact on the production level of the farmer, number of members in the household, cost of rice production, productivity, income from other production activities, savings, and average risk frequency on rice farmers' risk attitudes ($P < 0.1$).

## 5. Conclusions and Policy Implications

This study evaluates the risk attitude and the factors affecting the risk attitude of rice farmers in the Mekong Delta, Vietnam, based on evidence from the An Giang and Hau Giang provinces. Results showed that the interviewed farmers who have a neutral risk attitude account for about 53%. There are 32% of the rice farmers in the sample who have a risk-preferred attitude, and 15% have a risk-averse attitude. An ordered probit regression model was applied to determine the factors affecting rice farmers' risk attitudes. The estimated results show that a group of households with increased income from rice production and using loans for rice production will reduce risk aversion. In contrast, the age, property value, and number of rice crops per year are factors that increase rice farmers' risk aversion.

The results of this study have confirmed that when income from rice production increases, farmers' willingness to use loans in production activities reduces their risk aversion. This indicates that supportive policies to increase income from rice production and improve the accessibility of credit, especially formal credit sources, are key in helping farmers expand their production activities and invest in options with higher return rates. Additionally, farmers were found to secure their current asset value to guarantee their livelihood and productivity. This suggests that the alternatives and/or financial policies to compensate for the losses in the uncertainty and risks and incentivize increase in the income of rice farmers, (e.g., price subsidies, stabilizing input prices, output prices, preferential loan programs at formal credit institutions, and/or rice-crop insurance programs) should be considered to call for the farmer's willingness to accept potential risks of utilizing the chance to invest in high potential profitability projects. These are then estimated to expand their production and business, thereby creating favorable conditions to diversify income sources and livelihood activities.

Notably, this study cannot find significant effects of farmers' willingness to participate in agricultural insurance on their attitudes towards risk. In connection with the study results, it puts forward the continuous research relative to the contribution of an incentive platform from policy makers and other related agents (i.e., insurance companies) to enhance farmers' willingness to participate in rice-crop insurance and share the losses from the uncertainties and challenging risks in the production process.

**Author Contributions:** This article is a collaborative work of the authors, who are the staff of the Department of Finance and Banking, School of Economics, Can Tho University, Vietnam. Accordingly, K.T.P.D. and P.D.K. are the main authors, who stated the research proposal, conceptual framework and methodology, reviewed and edited the original draft, and revised the manuscripts; N.T.B., and T.T.H.P. investigated the survey, and completed the validation and software analysis; P.H.N. contributed to the writing—original draft preparation. All authors have read and agreed to the published version of the manuscript.

**Funding:** This research was funded by the Vietnam Ministry of Education and Training, grant number B2020-TCT-06, hosted by Can Tho University, Vietnam.

**Institutional Review Board Statement:** Not applicable.

**Informed Consent Statement:** Not applicable.

**Data Availability Statement:** The data presented in this study are available on request from the corresponding author. The data are not publicly available due to the copyright which belongs to the funding administration.



**Acknowledgments:** The authors would like to thank the Local Authorities of Vietnam's Mekong Delta for their cooperation. We also thank the students who provided support during the collection of the survey data. We would like to thank the anonymous reviewers for the critical recommendations to improve the original manuscripts.

**Conflicts of Interest:** The authors declare no conflict of interest.

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
