# Peer review of "Factors Affecting Risk Attitude of Rice Farmers: Evidence from Vietnam’s Mekong Delta"

_jrfm, doi:10.3390/jrfm15070278_

Round 1

Reviewer 1 Report

1. Formulas should be numbered and quoted.

2. The risk literature review should be written in more detail.

3. It is recommended to cite some papers related to your topic of authors in Vietnam or JRFM.

4. The results of this method should be compared with the results of other methods (if possible, for example, prospect theory; utility theory) which will make the paper better.

Author Response

Dear Reviewer,

We would like to thank you for your critical comments. We have careful discussed and revised our manuscript as your suggested. Please see the attachment our rebuttal letter with the detail descriptions of our revisions progress.

Wish all the best to you.

Authors

Reviewer 2 Report

What is the scientific question that this study intends to address?

What is the innovation of this study?

Line228-295: What is the meaning of P>0.1 mentioned in this section? What does the P-value refer to?

The literature review was not exhaustive and critical depth was lacking.

The abstract lacks elaboration on the research background and limitations of existing research.

The introduction lacks elaboration on research methods and research contents.

This manuscript lacks a comparative discussion of the findings with those of similar studies.

This manuscript lacks research limitations and prospects for future research.

English needs to be improved, and it is recommended to ask native English speakers to revise.

This manuscript has obvious defects, the research is not deep enough, the research significance is not deep enough, etc., so it is recommended to reject the manuscript.

Author Response

Dear Reviewer,

We would like to thank you so much for your valuable comments. We have tried to revised as our best knowledge. We hope our revisions are clear to you and other audiences.

Sincerely yours,

Authors

Reviewer 3 Report

I am pleased to have the opportunity to review this research paper. This study attempted to explore the Factors Affecting To Risk Attitude Of Rice Farmers In Mekong Delta, Vietnam: Evidences from An Giang and Hau Giang Provinces. Although the topic of this research study is interesting and fits within the journal scope, I think authors should apply the comments indicated below to increase the quality of research justification, contributions and findings. The manuscript know lacks in scientific style and structure.

First of all, paper research gap. Please improve this part in introduction section. Introduction is very general and lacked alignment to the research findings, no discussion was provided to derive the implication from. Theoretical and pragmatics implication are vague and need to be better aligned with this paper theoretical underpinnings and proposed process. Furthermore, there is insufficient support and weak arguments in support of the objective that is proposed as well as the model developed. In the final part of the introduction the objectives proposed, originality and gap that would be better covered. Also how the author will perform the methodology.

the topic of this research study is interesting and fits within the journal scope, I think authors should apply the comments indicated to increase the quality of research justification, contributions and findings

What is the originality of this research?  Paper research gap and originality should be better presented at the end of introduction section

Please consider this structure for manuscript final part.

-Discussion

-Conclusion

-Managerial Implication

-Practical/Social Implications

-Discussion needs to be a coherent and cohesive set of arguments that take us beyond this study in particular, and help us see the relevance of what authors have proposed. Authors should create an independent “Discussion” section. Author need to contextualize the findings in the literature, and need to be explicit about the added value of your study towards that literature. Also other studies should be cited to increase the theoretical background of each of the method used. Findings should be contextualized in the literature and should be explicit about the added value of the study towards the literature. Limitations and future research

Questions to be answered:

What practical/professional and academic consequences will this study have for the future of scientific literature (theoretical contributions)?

Why is this study necessary? should make clear arguments to explain what is the originality and value of the proposed model. This should be stated in the final paragraphs of introduction and conclusion sections.

Author Response

Dear Reviewer,

We would like to thank you so much for your valuable comments. We have tried to revised as our best knowledge. We hope our revisions are clear to you and other audiences. Please see the attachment to find the details of our revisions.

Sincerely yours,

Authors

Round 2

Reviewer 2 Report

The author carefully revised the manuscript. The current version is acceptable.

Author Response

Dear Reviewer,

We would like to thank you so much for your kind recommendations.

We received the 2nd round of revision requests of our manuscript. We have careful considered, and would like to response one by one your comments as follows:

Reviewer's comment:

1. Introduction part can be improved.

Our response: we have added more information in the manuscript to explain carefully why the study on risk attitudes is necessary for rice production in Vietnam. 

2.  All the cited references relevant to the research: can be improved

Our responses: we have double-checked the references lists to ensure all citation has been included in the reference lists and vice versa. Also, we updated the current relevant citation and references.

3. Comments and Suggestions: The author carefully revised the manuscript. The current version is acceptable.

Our responses: we would to thank you so much for your critical recommendations to improve our manuscript. 

We hope that you are satisfied to our revisions.

Wish all the best to you.

Authors

Reviewer 3 Report

The work is now much better, congratulations. I only ask you to justify in a more sustained way the need to carry out this study before be publlish

Author Response

Dear Reviewer,

We would like to thank you so much for your kind recommendations.

We received the 2nd round of revision requests of our manuscript. We have careful considered, and would like to response one by one your comments as follows:

Reviewer's comment:

The work is now much better, congratulations. I only ask you to justify in a more sustained way the need to carry out this study before be published.

Our response:

- We have added more information in the manuscript to explain carefully why the study on risk attitudes is necessary for rice production in Vietnam.

- We have double-checked the references lists to ensure all citation has been included in the reference lists and vice versa. Also, we updated the current relevant citation and references.

We really hope that you are satisfied to our revisions. We would to thank you so much for your critical recommendations to improve our manuscript.

Wish all the best to you.

Authors

This manuscript is a resubmission of an earlier submission. The following is a list of the peer review reports and author responses from that submission.